# A Sinusoidal Current Generator IC with 0.04% THD for Bio-Impedance Spectroscopy Using a Digital ΔΣ Modulator and FIR Filter

Soohyun Yun 🆔 and Joonsung Bae *

Department of Electrical and Electronic Engineering, Kangwon National University, Chuncheon 24391, Republic of Korea; the2706@kangwon.ac.kr
* Correspondence: baej@kangwon.ac.kr

**Abstract:** This paper presents a highly efficient, low-power, compact mixed-signal sinusoidal current generator (CG) integrated circuit (IC) designed for bioelectrical impedance spectroscopy (BIS) with low total harmonic distortion (THD). The proposed system employs a 9-bit sine wave lookup table (LUT) which is simplified to a 4-bit data stream through a third-order digital delta–sigma modulator (ΔΣM). Unlike conventional analog low-pass filters (LPF), which statically limit bandwidth, the finite impulse response (FIR) filter attenuates high-frequency noise according to the operating frequency, allowing the frequency range of the sinusoidal signal to vary. Additionally, the output of the FIR filter is applied to a 6-bit capacitive digital-to-analog converter (CDAC) with data-weighted averaging (DWA), enabling dynamic capacitor matching and seamless interfacing. The sinusoidal CG IC, fabricated using a 65 nm CMOS process, produces a 5 µA amplitude and operates over a wide frequency range of 0.6 to 20 kHz. This highly synthesizable CG achieves a THD of 0.04%, consumes 19.2 µW of power, and occupies an area of 0.0798 mm$^2$. These attributes make the CG IC highly suitable for compact, low-power bio-impedance applications.

**Keywords:** sinusoidal current generator; bio-impedance spectroscopy; delta–sigma modulator; impedance; electrochemical impedance spectroscopy; high linearity





## 1. Introduction

The bioelectrical impedance spectroscopy (BIS) technique [1] is a non-invasive method that uses an alternating current to measure the electrical properties of biological tissues, enabling the assessment of various physiological parameters, including body composition and cell health. The frequency range plays a crucial role in extracting relevant information from biological tissues. For instance, the β-dispersion range (below 100 kHz) is used to analyze tissue composition and membrane integrity for assessing body composition, while the α-dispersion range (below 1 kHz) is used to evaluate membrane characteristics crucial for cell health analysis. To effectively utilize the BIS technique across these diverse frequency ranges, it is essential to employ a current wave with a highly linear frequency tone to minimize harmonic distortions, along with a highly tunable frequency to accommodate a broad spectrum of BIS applications.

Figure 1 illustrates the impedance derivation process in a typical bioelectrical impedance spectroscopy (BIS) technique. As shown in Figure 1a, the overall system consists of a current generator (CG), an instrumentation amplifier (IA), and a chopper. The CG generates a current, $I_Z$, which flows through the unknown impedance, $Z_M$, which we aim to measure. The voltage across $Z_M$ is then applied to the input of the IA, denoted as $V_{IN}$. The IA amplifies this voltage by its gain, resulting in an output signal, $V_Z$. The chopper subsequently converts $V_Z$ to the DC band to determine its magnitude. The estimated value of the unknown impedance, $\widehat{Z_M}$, can be obtained by dividing the output voltage, $V_{OUT}$, by the applied current and the gain of the IA.

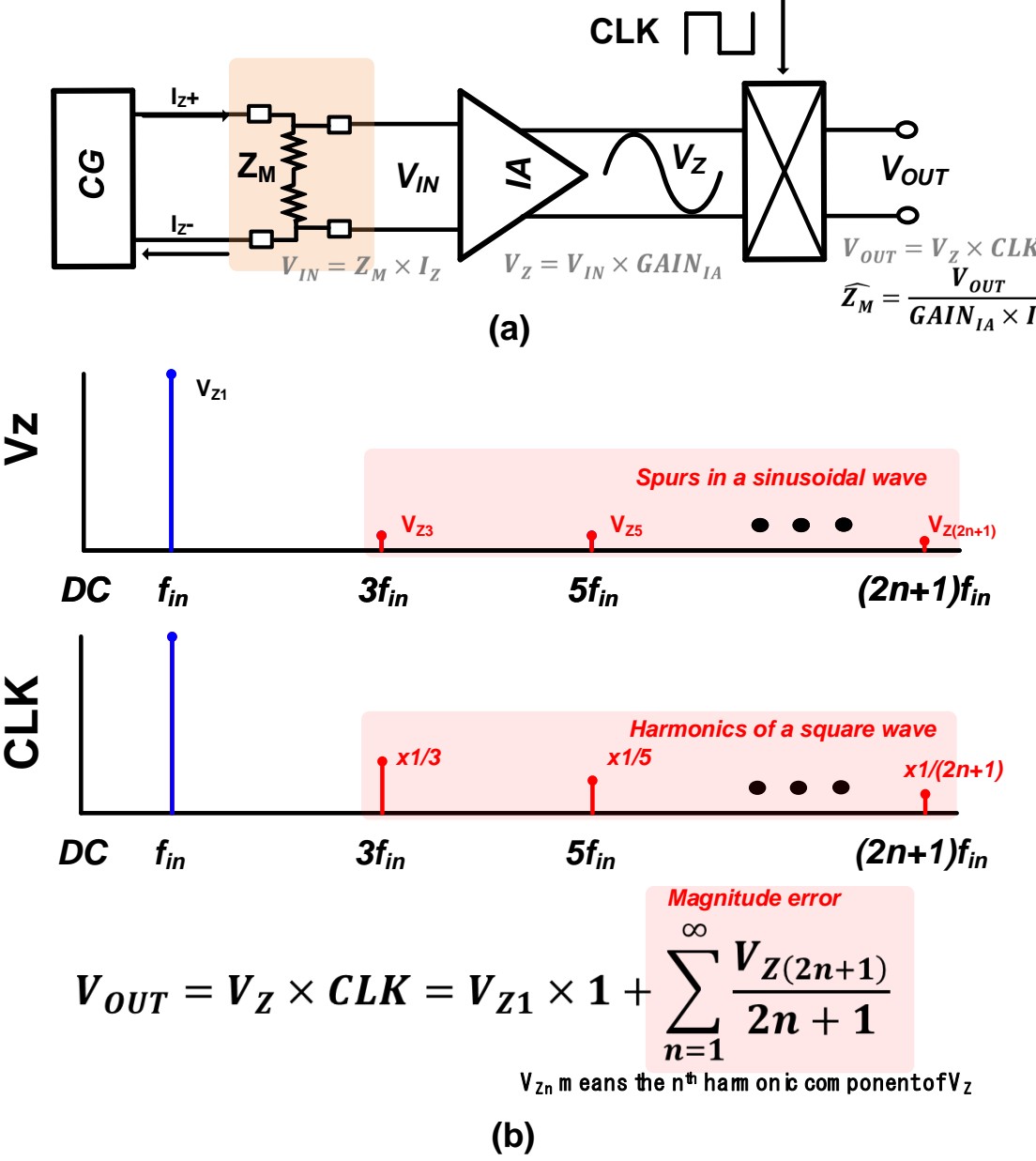

**Figure 1.** Error caused by the signal harmonic folding at the frequency mixing stage in the demodulator. (**a**) The overall system of BIS. (**b**) The effect of square waves on output voltage.

However, the process of obtaining $V_{OUT}$ introduces errors due to the multiplication of the IA output voltage, $V_Z$, with the square wave of the chopper (Figure 1b). The square wave comprises odd harmonic components with amplitudes inversely proportional to the harmonic order (1/n). When these components multiply with the harmonic components of the $V_Z$, the lower-order harmonics of the $V_Z$ exhibit non-negligible amplitudes in the DC band, leading to errors in the estimated impedance.

The accuracy of the estimated impedance, $\widehat{Z_M}$, is significantly degraded by the presence of harmonic components, with the third harmonic being the most dominant contributor to the error, as it is the odd harmonic component closest to the fundamental frequency. Since low-order harmonics have the greatest impact on the overall total harmonic distortion (THD) performance, minimizing the harmonic content of the current, $I_Z$, produced by the CG is crucial for achieving accurate impedance magnitude measurements. Additionally, the linearity of the IA plays a vital role in preserving the accuracy of the impedance estimation.

Several research approaches exist for implementing current wave generation [2]. One common approach is the use of square waves [3–6]. Although generating square waves is straightforward and power-efficient, this method introduces harmonics of significant magnitude, which complicates the extraction of accurate impedance measurements. Consequently, additional error correction techniques are required to mitigate the effects of these harmonics [6]. Because of their inherent inaccuracy, due to significant harmonic components, square-wave-based methods can be less reliable for precise impedance analysis.

Another approach employs sine wave oscillators [7–10], which generate signals with higher linearity compared to square waves. This results in fewer harmonic distortions, enhancing the accuracy of impedance measurements. However, oscillator-based methods have limited adaptability to the wide frequency ranges required for BIS applications. This can restrict their effectiveness across various types of bioelectrical impedance analyses.

Digital-to-analog converter (DAC)-based sinusoidal generation has gained significant popularity in recent years. This technique employs a sampling frequency higher than the target output frequency to produce pseudo-sinusoidal waveforms. DAC-based methods are particularly well-suited for BIS applications, as they cover frequencies spanning from the α-dispersion to the β-dispersion ranges, making them ideal for on-chip sinusoidal signal generation implementations. These methods are favored due to their ability to deliver sinusoidal waves with high linearity and tunable frequency control. Ongoing research [11–15] focuses on optimizing frequency scalability and reducing power consumption in mixed-signal systems by employing sine wave LUTs and DACs to generate sine waveforms with low THD, as illustrated in Figure 2.

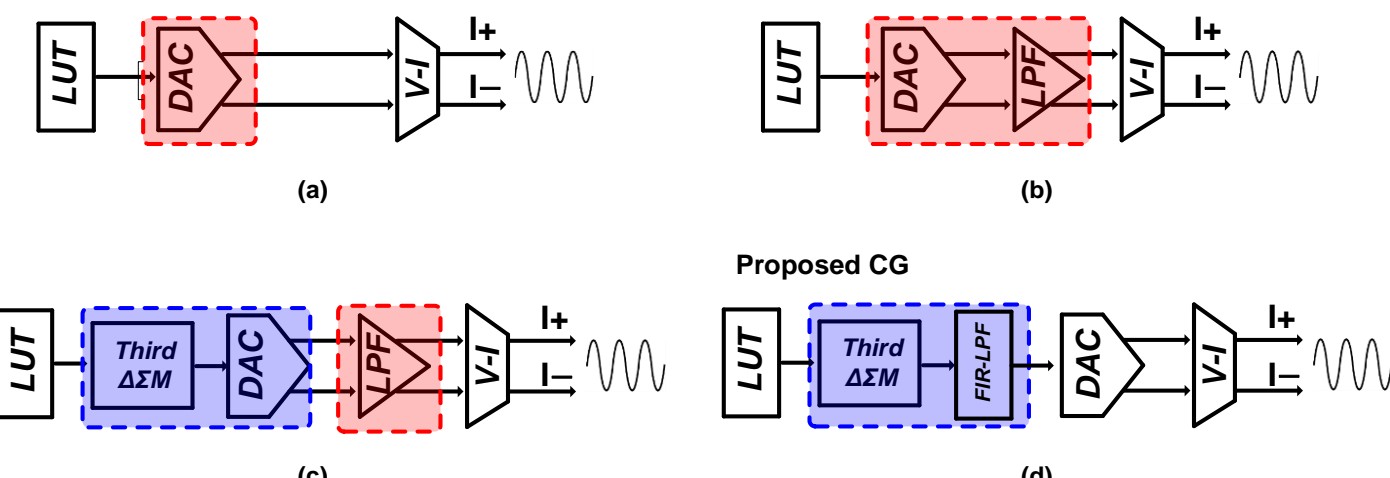

**Figure 2.** Conventional sinusoidal current generator IC. (**a**) LUT based CG. (**b**) LUT-LPF based CG. (**c**) LUT-LPF based ΔΣM CG. (**d**) Proposed ΔΣM CG.

In Figure 2a, the LUT is directly connected to a DAC to generate a current wave [11,14]. This configuration requires a high-resolution DAC to generate a sinusoidal wave with high linearity. However, high-resolution DACs are prone to mismatches and tend to require a significant area. The structure in Figure 2b adds an LPF to the LUT [12,13], enhancing linearity while reducing the demands on the DAC. This method still necessitates a high-resolution DAC to satisfy the linearity requirements and employs an analog LPF, which requires resistor and capacitor arrays to adjust the bandwidth of the LPF, leading to substantial area consumption. Figure 2c features a digital ΔΣM between the LUT and DAC to alleviate the DAC's linearity requirements, allowing the use of a low-resolution DAC to generate a highly linear current wave [15]. Moreover, a gm-C-type LPF can be utilized to achieve a highly linear current wave with minimal area use. However, this approach requires a higher-order analog LPF to eliminate high-frequency noise from the ΔΣM. Since the bandwidth of an analog LPF is not easily adjustable, this poses challenges in meeting the frequency scalability needs of BIS applications.

To address the limitations of existing bioelectrical impedance spectroscopy (BIS) technologies, such as frequency range constraints, high power consumption, and high total harmonic distortion (THD), we propose an innovative on-chip sinusoidal current generation application-specific integrated circuit (ASIC) for BIS applications. The structure of our ASIC, shown in Figure 2d, offers significant improvements in performance, power efficiency, and area optimization. The key features of our design include the following:

1.  Low power consumption: By employing a 0.5 V power supply and delta–sigma modulation ($\Delta\Sigma M$), the proposed design enables the use of a low-resolution digital-to-analog converter (DAC) to generate highly linear sinusoidal waveforms. This approach significantly reduces power consumption compared to existing solutions, making the ASIC more suitable for portable and wearable BIS applications.
2.  Wide frequency range with high linearity: The incorporation of a finite impulse response (FIR) filter, which acts as a low-pass filter (LPF), allows the ASIC to generate a wide range of frequencies while maintaining the high linearity of the sinusoidal output. This is achieved with a low area overhead, as the FIR filter eliminates the need for resistor and capacitor arrays, which are commonly used in existing designs.
3.  High dynamic range (DR): The ASIC utilizes a voltage-adjustable current DAC (CDAC) to generate currents of varying magnitudes. This feature facilitates a high dynamic range, enabling accurate impedance measurements across a wide range of biological tissues and conditions.

This paper is organized as follows: Section 2 introduces the proposed sinusoidal generator and its building blocks, including a third-order delta-sigma modulator, a 27th-order FIR filter, and a 6-bit CDAC. Section 3 presents the measurement results, followed by a discussion and suggestions for further work in Section 4. Finally, the paper is concluded in Section 5.

## 2. Proposed Sinusoidal Current Generator IC

Figure 3 presents a block diagram of the CG IC, which comprises a 9-bit LUT, a third-order $\Delta\Sigma M$, a 9-bit pseudo-random number generator (PRNG), an FIR filter, a data-weighted averaging (DWA) block, a differential 6-bit CDAC, and a voltage-to-current (V-I) converter. The current generator integrated circuit (CG IC) consists of several key components that work together to generate a highly linear, sinusoidal current output with low harmonic distortion. The process begins with a 9-bit look-up table (LUT) that contains the phase and amplitude information of a 9-bit sine wave. The LUT is designed to achieve a third-order harmonic distortion (HD) of less than $-70$ dB, ensuring a high-quality sinusoidal output. The 9-bit digital code from the LUT is then passed to a third-order delta–sigma modulator ($\Delta\Sigma M$), which converts it to a 4-bit digital code. To mitigate spurs caused by periodicity and maintain high linearity, the 4-bit code is dithered with a 9-bit pseudorandom number generator (PRNG). This dithering process effectively randomizes the quantization noise, reducing the impact of periodic patterns on the output signal.

Next, the 4-bit digital code is processed by a 27th-order finite impulse response (FIR) filter, which upsamples the signal and converts it to a 6-bit digital code. The FIR filter acts as a low-pass filter, suppressing high-frequency noise and unwanted harmonics introduced by the delta–sigma modulator. The high order of the FIR filter enables steep suppression of these unwanted components, contributing to the overall reduction in total harmonic distortion (THD). The 6-bit digital code is then passed to a data weighted averaging (DWA) block, which encodes the data into a random 61-bit thermometer code using a dynamic element matching scheme. This process helps minimize the impact of mismatches in the subsequent digital-to-analog conversion stage, further improving the linearity of the output signal. The 61-bit thermometer code serves as the input to a voltage-adjustable 6-bit differential current digital-to-analog converter (CDAC). The CDAC generates an analog AC voltage with a peak-to-peak swing determined by the difference between the positive and negative reference voltages (VREFP-VREFN). The adjustable nature of the CDAC allows for the generation of voltages with varying amplitudes, providing flexibility in the output signal. Finally, the output of the voltage-regulated CDAC is fed into a voltage-to-current (V-I) converter. This stage

takes the voltage signal and converts it into a highly linear, sinusoidal current output. The V-I converter is designed to be current-adjustable, enabling the CG IC to generate current signals with different magnitudes to suit various application requirements.

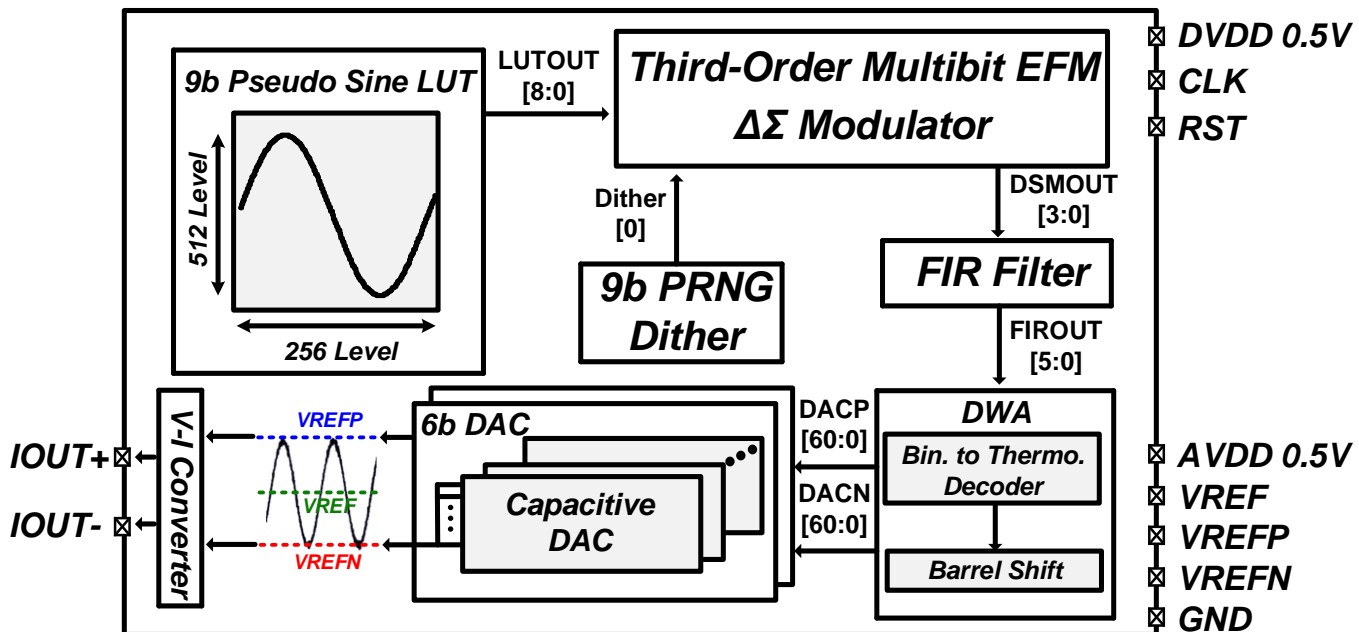

**Figure 3.** Block diagram of overall system.

### 2.1. Third-Order Delta–Sigma Modulator (ΔΣM)

Figure 4 shows the architecture of the third-order multi-stage noise shaping (MASH) ΔΣM structure with an oversampling ratio (OSR) of 256. The ΔΣM employs an error feedback modulator (EFM) topology, in which the first stage utilizes two output bits to reduce the quantization error in the overall output. This design choice also contributes to the reduction in the size of the second-stage and third-stage EFMs and their associated registers. By implementing this approach, the dynamic range of the ΔΣM is increased without consuming additional power, thereby improving the overall efficiency of the system.

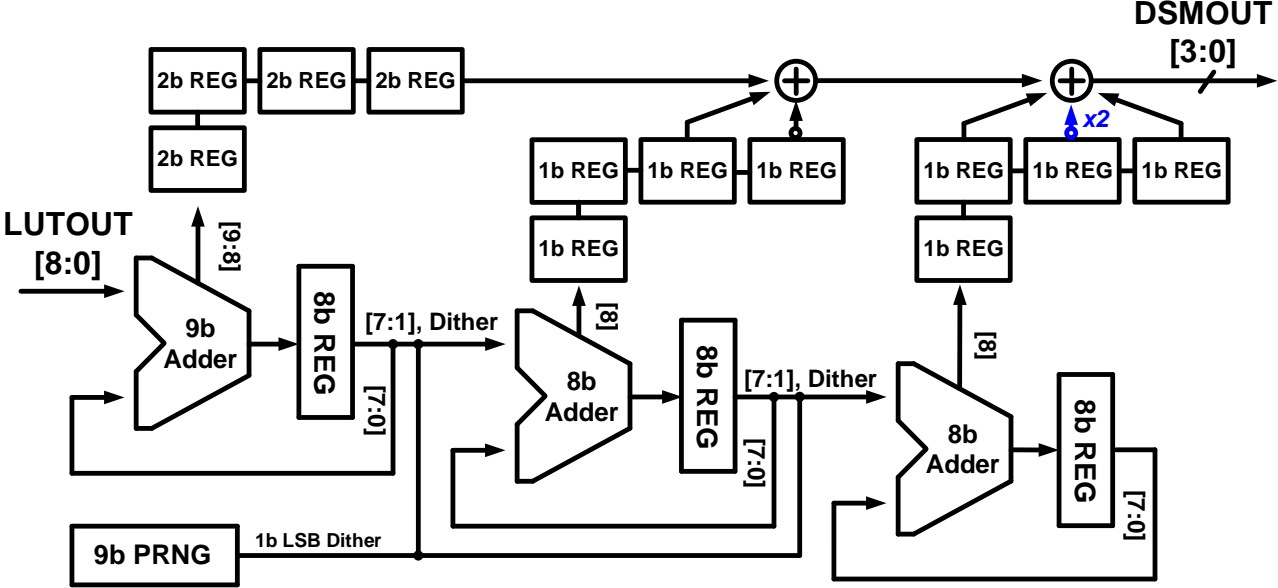

**Figure 4.** Third-order ΔΣM architecture.

To further enhance the linearity of the ΔΣM, a 9-bit PRNG is employed as the input to the least significant bit (LSB) of the second and third EFMs. This technique helps minimize errors caused by periodicity in the modulator [16]. The PRNG effectively randomizes the quantization noise, preventing the formation of tonal behavior and improving the spectral purity of the output signal.

The output of the third-order MASH ΔΣM is a 4-bit signal, which theoretically allows for a maximum value of 15 (4b'1111). However, due to practical limitations in the implementation, the maximum achievable value is limited to 9 (4b'1001). This constraint arises from the specific design choices made in the EFM stages and the overall architecture of the ΔΣM. Despite this limitation, the third-order MASH ΔΣM provides a high-resolution output with improved linearity and noise-shaping characteristics.

### 2.2. The 27th-Order Finite Impulse Response (FIR) Filter

Figure 5 presents the architecture of the 27th-order FIR filter employed in the CG IC. The FIR suppresses the overall spurs and provides consistent filtering performance across different frequency ranges. To minimize complexity, the FIR filter is implemented using a moving average filter structure with all coefficients set to 1. The order of the FIR filter is a critical design parameter that balances the suppression of harmonic components with the preservation of the fundamental frequency components. Using the periodic signal output from the third-order delta–sigma modulator (ΔΣM) as input, which is based on the look-up table (LUT) from the previous section, the filter order is chosen to ensure that the final output can be represented within a 6-bit range. Through analysis and simulation, it was determined that a 27th-order FIR filter provides the optimal balance between harmonic suppression and maintaining the target fundamental frequency. This order enables the filter to attenuate unwanted harmonics, particularly from the 7th harmonic onwards, by more than 10 dB, contributing significantly to the overall reduction in total harmonic distortion (THD). The 27th-order filter also ensures that the final output signal remains within the 6-bit range without causing overflow. While higher filter orders could provide even greater harmonic suppression, they would increase computational complexity and potentially introduce overflow issues. Therefore, the 27th-order FIR filter represents the optimal trade-off between effective THD reduction, preservation of the fundamental frequency, and compatibility with the 6-bit output resolution of the CG IC.

A 4-bit digital code input is essentially computationally expanded to a 9-bit output when passed through the 27th-order FIR filter. However, it is important to note that the input to the ΔΣM is a periodic 9-bit LUT component, and as previously mentioned, the ΔΣM output is limited to a maximum value of 9 (4b'1001). Since the input is periodic and does not utilize all 4 bits, using an output with fewer than 9 bits will not cause an overflow. By considering this characteristic, the DC component caused by the periodic signal can be removed during the FIR filtering process. This approach allows for the generation of a filtered output waveform using only 6 bits without encountering overflow issues. The 6-bit output is then converted to a thermometer code, which undergoes a barrel shift operation and is transformed into a randomized 61-bit code. This randomized code serves as the input to the DAC.

Figure 6 shows the output spectrum of the ΔΣM and the frequency response of the FIR filter. The FIR filter exhibits a suppression effect from the third harmonic component and attenuates the seventh harmonic and high-frequency noise by more than 10 dB. This filtering effectively suppresses the harmonic components of the linear sine wave and the high-frequency noise shaped by the ΔΣM. By attenuating these undesired components, the FIR filter contributes to the overall linearity and spectral purity of the generated sinusoidal signal.

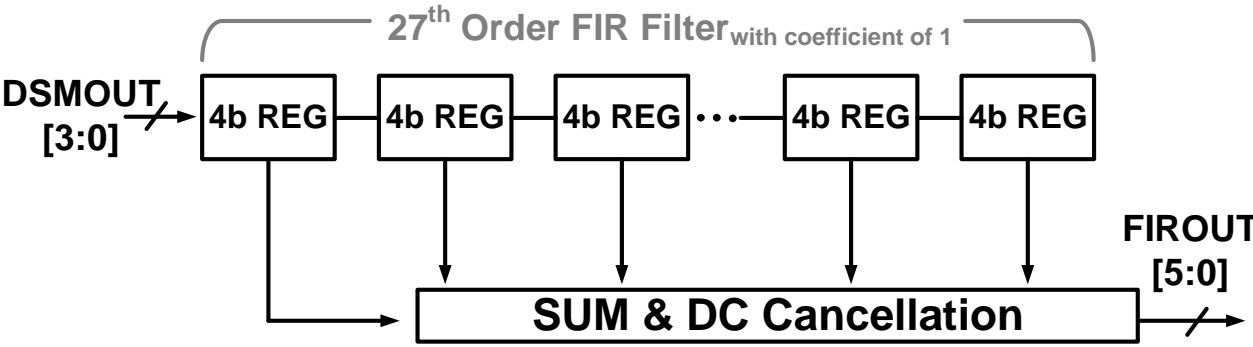

**Figure 5.** Implementation of a 27th-order FIR filter with a coefficient of 1.

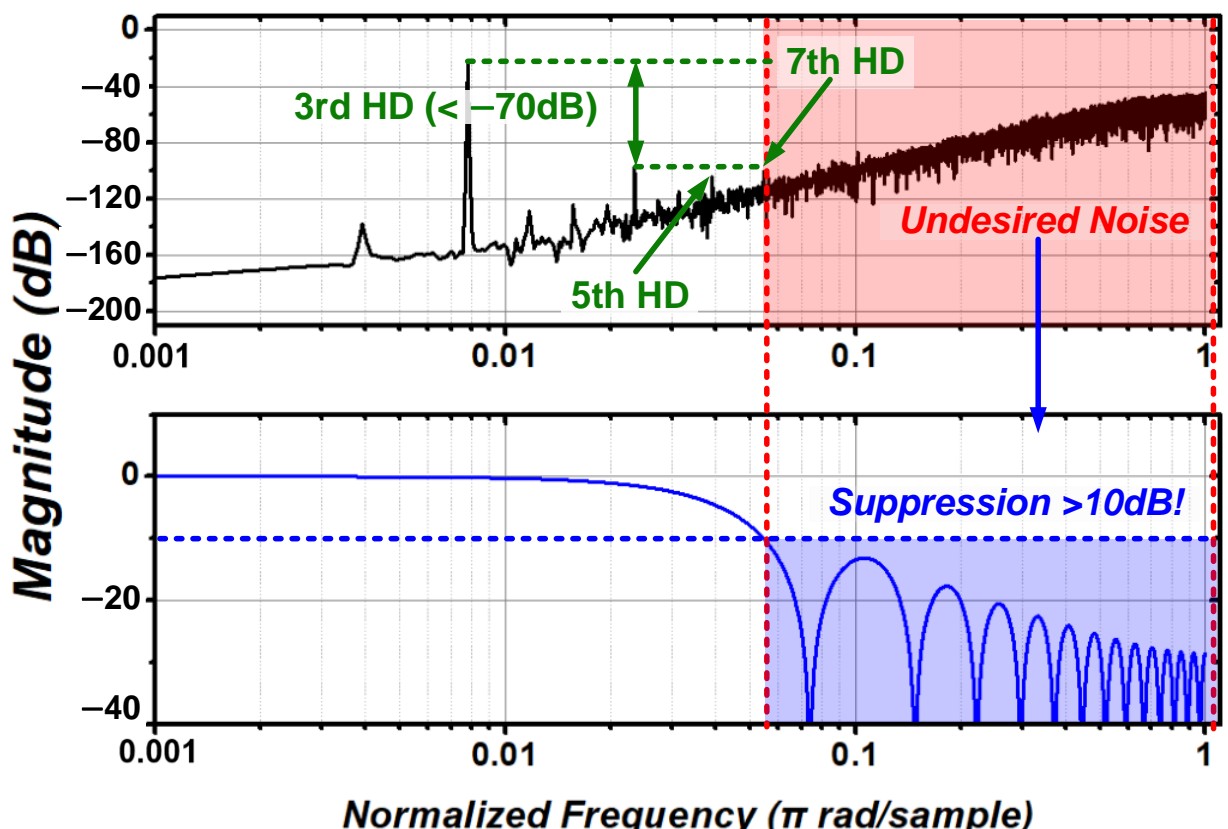

**Figure 6.** Output spectrum of ΔΣM and frequency response of FIR filter.

### 2.3. The 6-Bit Differential CDAC

Figure 7a illustrates the structure of the CDAC employed in the CG IC. The CDAC is designed to generate differential outputs, VOUT+ and VOUT−, and consists of a total of 122 cells, with 61 cells dedicated to each output. The inputs to the CDAC are the 61-bit thermometer codes DACP and DACN, which are obtained from the randomized DWA block connected to the output of the FIR filter. Additionally, the CDAC receives DC voltages VREF, VREFP, and VREFN. To generate the differential outputs, two CDACs are configured to produce the voltages VOUT+ and VOUT− using circuitry that inverts the inputs of DACP and DACN. The amplitude of the generated output voltages is controlled by the voltages VREFP and VREFN. These voltages can be utilized to adjust the output current amplitude of the V-I converter.

Figure 7b depicts the operating principle of the CDAC. Each CDAC cell is composed of four capacitors. The output voltage is generated by charging and discharging the capacitance using two capacitors for each half-cycle of the clock (CLK) signal. This approach

ensures high accuracy by minimizing the ripple voltage between the capacitors. By employing this technique, the CDAC maintains a stable and accurate output voltage, which is essential for the proper operation of the V-I converter and the overall performance of the CG IC. The use of a differential structure and the careful design and layout of the CDAC cells contribute to the high-linearity characteristics of the generated sinusoidal signal. The ability to control the output voltage amplitude through the regulation of VREFP and VREFN provides flexibility in adjusting the output current amplitude of the V-I converter, enabling the CG IC to accommodate various application requirements.

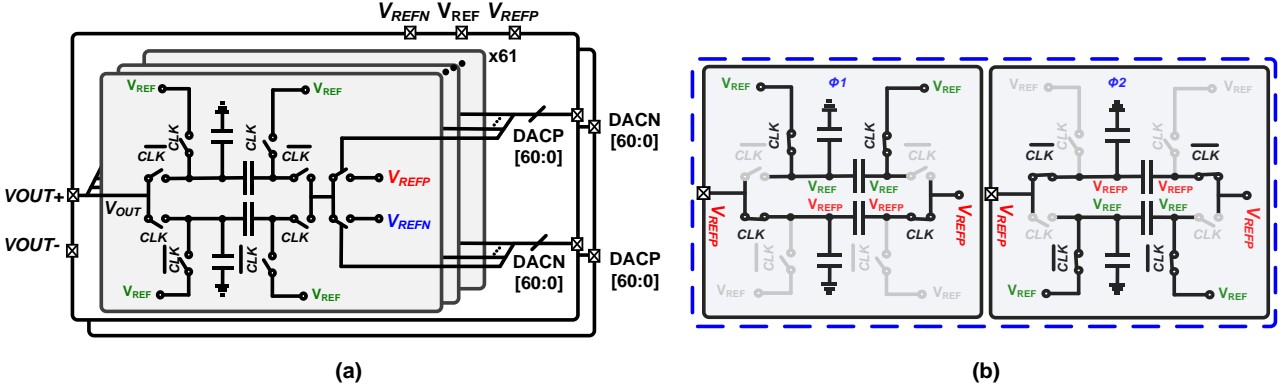

**Figure 7.** (**a**) Differential 6-bit CDAC. (**b**) Operation of CDAC.

## 3. Measurement Results

The CG IC is fabricated using a 65 nm CMOS process technology. Figure 8a presents a microphotograph of the fabricated chip, highlighting the compact size of the proposed CG IC, which occupies an active area of only 0.0798 mm$^2$.

Figure 8b illustrates the power breakdown of the CG IC when generating a 20 kHz sinusoidal wave with an amplitude of 5 μA. The CG IC operates with a supply voltage of 0.5 V and a clock frequency of 5.12 MHz. The power consumption of the digital block, which includes the ΔΣM, PRNG, FIR filter, and DWA, accounts for 60.4% of the total power consumed by the CG IC. This significant contribution of the digital block to the overall power consumption emphasizes the importance of optimizing the digital circuitry to achieve high energy efficiency.

The right side of Figure 8b provides a detailed breakdown of the power consumption within the digital block. Notably, the FIR filter emerges as the dominant power consumer, accounting for 52.57% of the digital block's power consumption. This observation highlights the critical role played by the FIR filter in the overall power budget of the CG IC.

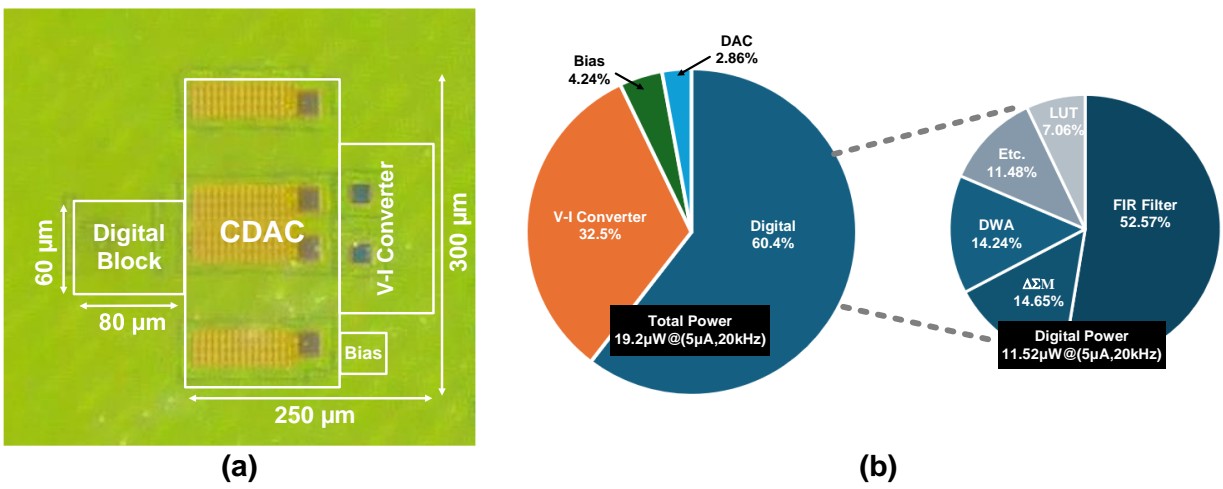

**Figure 8.** (**a**) Chip photograph. (**b**) Power breakdown of the CG IC.

Figure 9 presents the fast Fourier transform (FFT) analysis of the outputs from the LUT, ΔΣM, and FIR filter stages. The THD is calculated by summing the contributions of harmonics up to the 20th order. The THD values for the 9-bit LUT, ΔΣM, and FIR filter outputs are 0.048%, 0.096%, and 0.02%, respectively. Notably, the THD of the FIR filter output is more than two times better than that of the 9-bit LUT, demonstrating the effectiveness of the FIR filter in suppressing the harmonic components of the LUT output.

Figure 10 illustrates the FFT results at the output of the FIR filter and the analog output of the V-I converter. The FIR filter output represents the 6-bit code converted through the DAC, while the V-I converter output is measured at the output node of the V-I converter. The measured THD of the V-I converter output is approximately 0.04% due to the additional nonlinearity of the CDAC and V-I converter, while the THD of the FIR filter output is about 0.02%. Additionally, the V-I converter output achieves a spurious-free dynamic range (SFDR) of −71.2426 dBc.

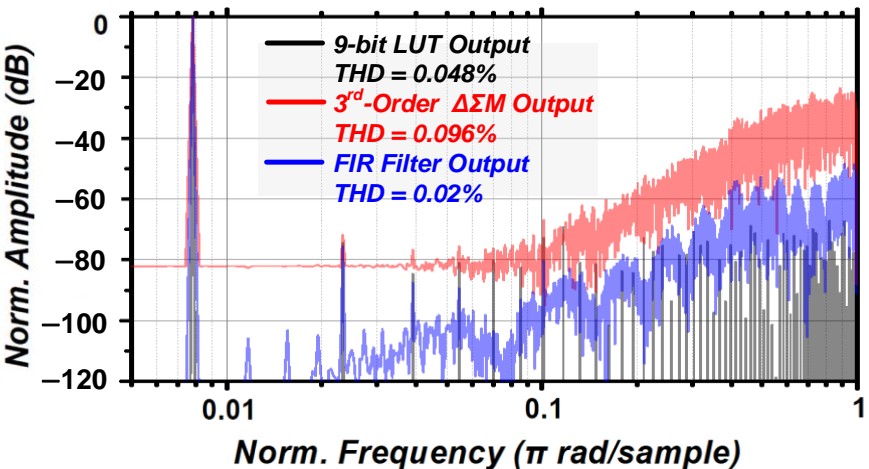

**Figure 9.** Comparison of the output of a 9-bit sinusoidal LUT, the output of a 4-bit third-order ΔΣM, and the output of a 6-bit FIR filter.

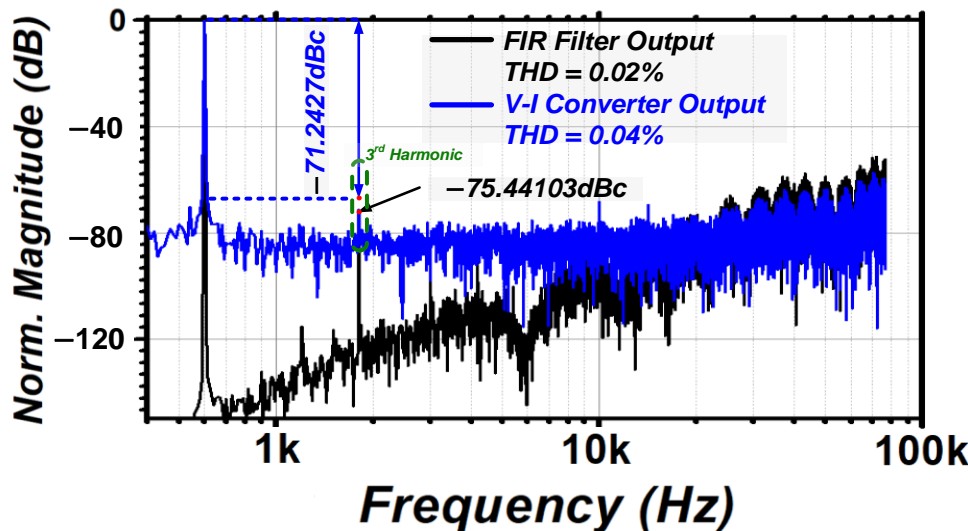

**Figure 10.** Comparison of FIR filter output and V-I converter output.

Figure 11a depicts the measured third-order HD and THD values as a function of the frequency of the generated current sinusoid. The CG IC covers a frequency range of 0.6 to 20 kHz, exhibiting a maximum THD of 0.04% and a minimum THD of 0.031% within the range. The third-order HD, which has the most significant impact on DC band demodulation, remains below −70 dB across the measured frequency range, consistent with the SFDR.

Figure 11b shows the output waveform when a 5 μA current is applied to a 2 kΩ resistor, representing the V-I converter output waveform previously shown in Figure 10.

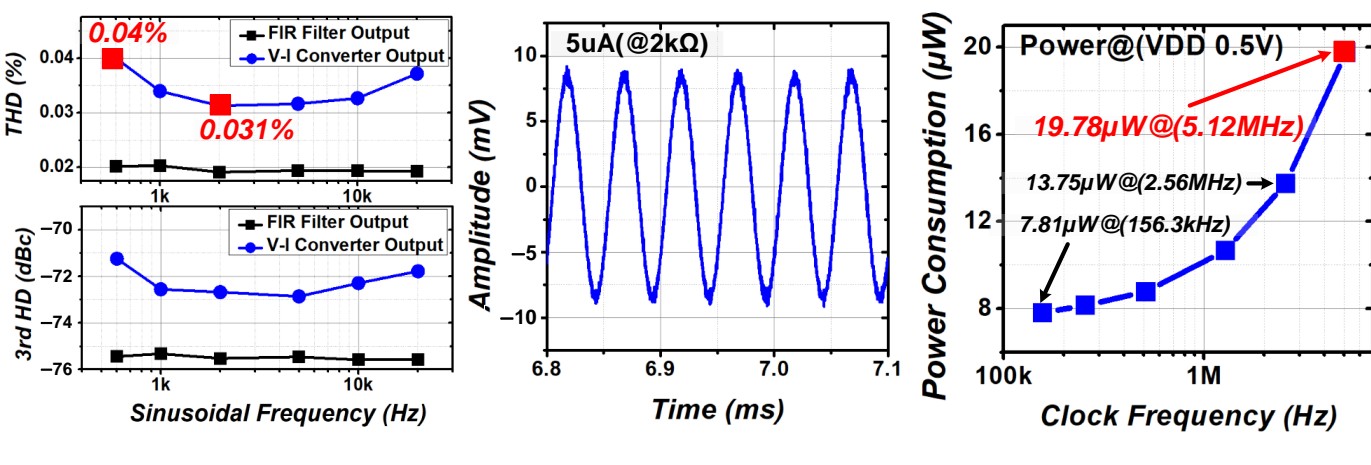

**Figure 11.** (**a**) THD and third-order HD as a function of its sinusoidal frequencies. (**b**) Analog waveform with 2 kΩ load at 20 kHz. (**c**) Power consumption as a function of clock frequency.

Figure 11c presents the measured power consumption of the CG IC over a range of operating frequencies from 153.6 kHz to 5.12 MHz, corresponding to sinusoidal output frequencies of 0.6 kHz and 20 kHz, respectively. The power consumption is measured while generating a current sinusoid with an amplitude of 5 μA. At an operating clock frequency of 156.3 kHz, the CG IC consumes 7.81 μW for a 0.6 kHz sinusoidal output. In contrast, at a clock frequency of 5.12 MHz, the power consumption increases to 19.2 μW for a 20 kHz sinusoidal output.

Table 1 presents a performance comparison between the proposed CG IC and conventional CG ICs for bio-impedance (bio-Z) sensing applications. We've highlighted this work. The proposed architecture, which combines a discrete-time finite impulse response (DT-FIR) filter with a delta–sigma modulator (ΔΣM), enables the generation of sinusoidal signals from 0.6 to 20 kHz with a total harmonic distortion (THD) below 0.04%. Compared to [15], the proposed design employs a higher oversampling ratio (OSR) and generates a 5 μA sinusoidal current, resulting in increased power consumption. The use of FIR filters in the digital block also leads to a larger area. However, these design choices enable frequency adjustability and superior THD performance. The FIR filters effectively suppress harmonic components, ensuring a consistent, low-distortion sinusoidal output across the entire frequency range. Despite the trade-offs in power and area, the ability to adjust the operating frequency and maintain low distortion levels sets our design apart, offering significant advantages in flexibility and signal quality. The wide frequency coverage and low THD achieved by the proposed CG IC make it an attractive solution for accurate and reliable bio-Z sensing applications.

**Table 1.** Performance summary and comparison table.

|  | **This Work** | TCAS-II 2019 [17] | TBCAS2019 [12] | JSSC20 [14] | JSSC22 [15] | ESSCIRC 23 [13] |
|---|---|---|---|---|---|---|
| Technology (nm) | **65 nm** | 180 nm | 130 nm | 65 nm | 65 nm | 180 nm |
| Supply (V) | **0.5** | 1.2 | 1 | 0.5 | 0.5 | 1.2 |
| Architecture | **LUT + DT-FIR** | RDO + DT-FIR | LUT + CT-LPF | LUT | LUT + CT-LPF | LUT + CT-LPF |
| Injection Frequency (kHz) | **0.6–20** | 20 | 15–125 | 20 | 20 | 0.1–30 |
| Injection Amplitude (μA) | **1–5** | 40–160 | 20–200 | 8.4 | 2 | 10–100 |
| THD (%) | **0.04** | 0.17 | −58 dBc (SFDR) | 0.66 | 0.088 | 0.4 |
| CG Power (μW) | **19.2@5 μA** | 55.6@40 μA | 538.56@100 μA | 5.31@8.4 μA | 6.2@2 μA | N.R |
| CG Area (mm$^2$) | **0.0798** | 0.052 | 0.436 | 1.958 | 0.059 | N.R |

## 4. Discussion with Further Work

The current system employs a $\Delta\Sigma$M to generate sinusoidal waves, necessitating a high operating frequency to produce high-frequency sinusoids. Consequently, miniaturizing the OSR is crucial for achieving a wider frequency range. Furthermore, the code-to-signal conversion in the digital-to-analog domain is implemented using a CDAC in the voltage domain, requiring an additional V-I converter to generate current waveforms. As illustrated in Figure 8b, the V-I converter consumes a significant portion (32.5%) of the total power, highlighting the need for power optimization in this stage.

Moreover, the linearity of the generated current waveform is a concern. Figure 10 reveals that the THD of the current waveform generated by the V-I converter, considering harmonics up to the 20th order, ranges from 0.02% to 0.04%. This indicates suboptimal linearity performance, which may impact the accuracy of impedance measurements. Additionally, the absence of a readout circuit in the current system hinders the verification of the impedance measurement results, limiting the assessment of the system's overall performance. To address these challenges and enhance the performance of current-mode ICs for bioimpedance applications, future work should focus on reducing the OSR while adopting low-bit current DACs. By directly generating current waveforms using current DACs, the need for a separate V-I converter can be eliminated, potentially leading to power savings and improved linearity.

In order to improve the accuracy of impedance measurement and verify the generated current waveforms, it is essential to design a readout circuit that satisfies low power, low area, and high linearity. Therefore, the inclusion of a readout circuit in future work is essential for verifying the accuracy of impedance measurements. However, the design of this readout circuit must carefully consider the constraints and performance requirements of the overall system. As current systems, including our CG, prioritize low power consumption and small area, the readout circuit must also be designed with these aspects in mind to ensure compatibility and maintain the overall system efficiency. A notable example of a low-power readout circuit is presented in [14]. This circuit operates with a 0.5 V drive voltage and consumes only 3.95 μW of power while achieving a low noise level of 45 nV/$\sqrt{\text{Hz}}$ and a bandwidth of 408 kHz, which helps minimize phase error. However, it is important to note that the THD performance of the input current wave in [14] was 0.66%, which is higher than the current wave THD achieved by the proposed CG. To ensure the accuracy of impedance measurements, it is crucial to implement the readout circuit with high linearity to avoid compromising the linearity of the generated current. The readout circuit's linearity should be at least as good as the linearity of the CG to maintain the overall system performance. In our future work, the most critical aspect of the readout circuit design will be to achieve a THD of less than 0.04% when inputting a pure sine wave that satisfies the targeted input amplitude. This stringent THD requirement must be met while simultaneously satisfying the low power and low area constraints. Achieving this level of linearity in the readout circuit will be challenging, as it requires careful design considerations and trade-offs between power consumption, area, and performance.

Commercial electrical impedance spectroscopy (EIS) systems, such as [18], employ single-tone excitation methods that enable the measurement of a wide range of impedances with high accuracy. However, these systems often face limitations in terms of portability and power efficiency. Their large size makes them difficult to transport and use in various settings, while their high power consumption necessitates the use of an external power supply, further restricting their mobility and ease of use.

In contrast, the proposed CG IC, when integrated with a readout circuit, offers a compelling solution for miniaturized and power-efficient BIS systems. With its small area of 0.0798 mm$^2$ and low power consumption of 19.2 μW, our design enables the development of compact and energy-efficient BIS devices that can be easily integrated into portable or implantable systems. The miniaturization and low-power characteristics of the CG IC make it particularly well-suited for applications that require continuous monitoring or on-the-go measurements. In the field of medical diagnostics, the design can be incorporated into

portable devices for point-of-care testing, allowing healthcare professionals to perform rapid and accurate assessments of various physiological parameters in a variety of settings, including clinics, hospitals, and even remote locations. Furthermore, the small size and low power consumption of the CG IC open up new possibilities for wearable and implantable BIS devices. These devices can be used for continuous monitoring of body composition changes, providing valuable insights into an individual's health status over time. The miniaturized nature of our design ensures patient comfort and enables long-term monitoring without the need for frequent device replacement or recharging. The integration of the CG IC with a readout circuit also offers the potential for a wide range of other applications beyond medical diagnostics and body composition analysis. For example, it can be used in environmental monitoring systems to assess the quality of water or soil, or in industrial settings to monitor the composition and stability of various materials.

In summary, the small area, low power consumption, high linearity, and frequency tunability of our proposed CG IC, when combined with a readout circuit, provide a comprehensive solution for miniaturized and power-efficient BIS systems. This innovative design overcomes the limitations of commercial EIS systems and enables the development of portable, wearable, and implantable devices that can revolutionize the way we approach impedance spectroscopy in various applications, including medical diagnostics, body composition analysis, and beyond.

### 5. Conclusions

In conclusion, this paper presents a highly efficient and compact sinusoidal CG IC architecture suitable for BIS applications. The proposed design leverages a third-order $\Delta\Sigma M$ in combination with DT-FIR filtering to generate sinusoidal waves with high linearity across a wide range of frequencies. The IC incorporates a third-order $\Delta\Sigma M$, an FIR filter, and a DWA technique to mitigate the effects of DAC burden and mismatch, while employing a 6-bit CDAC for implementation. Experimental results demonstrate that the CG IC, operating with an OSR of 256, is capable of generating sinusoidal waves with remarkable linearity, achieving a THD below 0.04% within the frequency range of 0.6 to 20 kHz. This demonstrates the potential of the proposed architecture as a highly scalable and synthesizable sinusoidal CG solution for compact BIS applications in various fields, such as medical diagnostics, body composition analysis, and tissue characterization.

**Author Contributions:** Conceptualization, S.Y. and J.B.; Methodology, S.Y.; Writing—original draft, S.Y.; Writing—review and editing, J.B. All authors have read and agreed to the published version of the manuscript.

**Funding:** This work was supported by a research grant from the Kangwon National University in 2024, the Korea Institute for Advancement of Technology (KIAT) grant, funded by the Korea Government (MOTIE) (P0017011 and P0020966), HRD Program for Industrial Innovation, and the Technology Innovation Program under Grant 20012355 (Fully Implantable Closed Loop Brain to X for Voice Communication), funded by the Ministry of Trade, Industry & Energy (MOTIE).

**Data Availability Statement:** Data are contained within the article.

**Conflicts of Interest:** The authors declare no conflicts of interest.

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
