# Peer review of "A Sinusoidal Current Generator IC with 0.04% THD for Bio-Impedance Spectroscopy Using a Digital ΔΣ Modulator and FIR Filter"

_electronics, doi:10.3390/electronics13224450_

Round 1
Reviewer 1 Report
Comments and Suggestions for Authors
The Authors presented a novel architecture of sinusoidal current generator IC with high linearity. The circuit has been designed, measured and compared to the technical literature. The results are well-presented, well-written and of interest to the Reader. In the discussion we suggest to better highlight pros and cons compared to to previous results. In particular:
- TDH performance is said to be better than [15], but the area and power consumption are both larger: can you comment on this?
- In the discussion, the presence of the readout circuit is presented as an important feature for future developments: is it present in circuits mentioned in Table 1? If so, this needs to be highlighted at the end of Section 3 for better comparison in terms of area and power cunsumption.
Author Response
Comments 1 : TDH performance is said to be better than [15], but the area and power consumption are both larger: can you comment on this?:
Response 1 : We appreciate your observation regarding the power consumption and area of our design compared to [15]. Our design employs a higher oversampling ratio (OSR) and generates a 5uA current wave, which contributes to increased power consumption. Additionally, the use of FIR filters in the digital block results in a larger area. However, these design choices enable our system to operate at different frequencies and achieve superior THD performance. We have addressed this point and emphasized the trade-offs between performance, power, and area on lines L.321-329 of the revised manuscript.
Key Revisions:
- Clarified the reasons for higher power consumption (higher OSR and 5uA current wave).
- Explained the larger area is due to the use of FIR filters in the digital block.
- Highlighted the benefits of the design choices: frequency adjustability and improved THD performance.
Comments 2 : In the discussion, the presence of the readout circuit is presented as an important feature for future developments: is it present in circuits mentioned in Table 1? If so, this needs to be highlighted at the end of Section 3 for better comparison in terms of area and power cunsumption.:
Response 2 : Thank you for raising this important point. To clarify, the circuits presented in Table 1 focus solely on the performance of the current generator (CG) and do not include the readout circuit. We acknowledge that the term "SSG Power" in the original table may have caused confusion, as it refers specifically to the power consumption of the CG when generating a specific current. To address this, we have updated the table, replacing "SSG Power" with "CG Power" to accurately reflect the scope of the comparison. This change has been emphasized in the revised manuscript.

Reviewer 2 Report
Comments and Suggestions for Authors
This paper presents a sinusoidal current generator integrated circuit (IC) specifically designed for bio-impedance spectroscopy (BIS), offering high efficiency, low power consumption, and a compact design. Detailed suggestions for improvements and optimization are provided below:
-
The authors explain the bioelectrical impedance spectroscopy technique, emphasizing the importance of the frequency range. However, they should also address the limitations of existing technologies, such as frequency range constraints, high power consumption, and high total harmonic distortion (THD) in current devices. Furthermore, the advantages of the proposed method in overcoming these issues should be clearly highlighted.
-
Figure 1 illustrates the impedance derivation process in a typical BIS technique. All symbols used in the figure need to be clearly explained, particularly in terms of how these parameters contribute to harmonic distortion and affect the overall system performance. A detailed explanation would improve understanding.
-
In Section 2, the authors present a diagram of the system setup, which includes the 3rd-order delta-sigma modulator, 27th-order finite impulse response (FIR) filter, and 6-bit differential CDAC. It would be beneficial to provide more detail on how these components interact within the CG IC, as well as the rationale for selecting a 27th-order FIR filter, and how it contributes to THD reduction.
-
For future work, the inclusion of a readout circuit would be advantageous for verifying the accuracy of impedance measurements. When discussing the addition of this circuit, the associated challenges and limitations should also be addressed.
-
The authors claim that the BIS IC can be applied in various fields, including medical diagnostics and body composition analysis. A more detailed discussion on the specific implementation of the design in these applications would strengthen the paper, providing clear insights into its practical uses.
Author Response
Comments 1 : The authors explain the bioelectrical impedance spectroscopy technique, emphasizing the importance of the frequency range. However, they should also address the limitations of existing technologies, such as frequency range constraints, high power consumption, and high total harmonic distortion (THD) in current devices. Furthermore, the advantages of the proposed method in overcoming these issues should be clearly highlighted.
Response 1 : We appreciate your valuable feedback and acknowledge the need to provide a more comprehensive discussion of the limitations of existing technologies and the advantages of our proposed method.
In the revised manuscript, we have expanded upon the challenges faced by current bioelectrical impedance spectroscopy devices, specifically addressing the following limitations:
- Frequency range constraints: Many existing devices have a limited frequency range, which hinders their ability to perform accurate and comprehensive impedance measurements across different biological tissues.
- High power consumption: Current devices often consume significant power, which can limit their applicability in portable or wearable systems and raise concerns about battery life and heat generation.
- High total harmonic distortion (THD): Existing technologies often suffer from high THD, which can degrade the accuracy and reliability of the impedance measurements, particularly at higher frequencies.
To highlight the advantages of our proposed method in overcoming these limitations, we have made the following additions to the manuscript in lines L.105-124:
- Low power consumption: By employing a 0.5V operating power, our design significantly reduces power consumption compared to existing solutions, making it more suitable for portable and wearable applications.
- Minimized THD: The use of delta-sigma modulation (DSM) and finite impulse response (FIR) filters in our design effectively minimizes THD, ensuring accurate and reliable impedance measurements across the entire frequency range.
- Low area implementation: The incorporation of FIR filters eliminates the need for resistor and capacitor arrays, enabling a low area implementation without compromising performance.
- Consistent filtering performance: Our design leverages FIR filters to maintain constant filtering performance regardless of the sampling frequency, ensuring robust and reliable operation across different measurement conditions.
Comments 2 : Figure 1 illustrates the impedance derivation process in a typical BIS technique. All symbols used in the figure need to be clearly explained, particularly in terms of how these parameters contribute to harmonic distortion and affect the overall system performance. A detailed explanation would improve understanding.
Response 2 : Thank you for highlighting the need for a clearer explanation of the parameters in Figure 1. We agree that a more detailed description of these parameters and their impact on harmonic distortion and system performance will enhance the reader's understanding of the BIS technique.
To address this, we have made the following revisions:
- Updated Figure 1: We have revised Figure 1 to include clear labels for all the parameters and components involved in the impedance derivation process. This visual improvement will help readers quickly identify and understand the role of each parameter in the BIS technique.
- Expanded explanation in the text (L. 37-59): We have added a more comprehensive explanation of the parameters in the text, focusing on their contributions to harmonic distortion and overall system performance.
Comments 3 : In Section 2, the authors present a diagram of the system setup, which includes the 3rd-order delta-sigma modulator, 27th-order finite impulse response (FIR) filter, and 6-bit differential CDAC. It would be beneficial to provide more detail on how these components interact within the CG IC, as well as the rationale for selecting a 27th-order FIR filter, and how it contributes to THD reduction.
Response 3 : Thank you for your suggestion to provide more clarity on the interaction between the key components within the current generator integrated circuit (CG IC) and the rationale behind selecting a 27th-order FIR filter for THD reduction.
We have made the following revisions to address your concerns:
- Detailed explanation of component interaction (L.136-164): We have expanded the description of how the 3rd-order delta-sigma modulator, 27th-order FIR filter, and 6-bit differential CDAC interact within the CG IC. The revised text now clearly explains that the delta-sigma modulator receives the input signal and generates a high-frequency, low-resolution output. This output is then fed into the FIR filter, which acts as a low-pass filter to remove the high-frequency noise and unwanted harmonics introduced by the modulator. The filtered signal is then passed to the 6-bit differential CDAC, which converts the digital signal to an analog current output while maintaining high linearity and dynamic range.
- Rationale for selecting a 27th-order FIR filter (L.192-207): We have added a detailed explanation of why we chose a 27th-order FIR filter for our design. The key reason is that higher-order filters have narrower passbands, enabling steeper suppression of unwanted frequency components. The 27th-order filter was found to be the maximum order that could be used without causing overflow in the final 6-bit output when using a periodic signal as input, while still maintaining the target fundamental frequency component. Additionally, this filter order enables suppression of more than 10 dB from the 7th harmonic onward, contributing to the achievement of THD results down to the 20th harmonic.
- Contribution to THD reduction: The revised text now emphasizes how the 27th-order FIR filter plays a crucial role in reducing THD. By effectively suppressing the unwanted harmonic components introduced by the delta-sigma modulator, particularly from the 7th harmonic onward, the filter ensures that the output signal has significantly reduced harmonic distortion. This, in turn, leads to improved overall THD performance of the CG IC.
Comments 4 : For future work, the inclusion of a readout circuit would be advantageous for verifying the accuracy of impedance measurements. When discussing the addition of this circuit, the associated challenges and limitations should also be addressed.
Response 4 : Thank you for highlighting the importance of discussing the challenges and limitations associated with the inclusion of a readout circuit in future work. We agree that addressing these aspects is crucial for providing a comprehensive perspective on the potential integration of the readout circuit with the current generator (CG) we have implemented.
In the revised manuscript, we have expanded on the discussion of the readout circuit and its implications for the overall system performance. Specifically, we have made the following additions:
- Low power and area considerations: We acknowledge that current systems, including our CG, are designed with a focus on low power consumption and small area. Therefore, the readout circuit must also be designed with these constraints in mind to ensure compatibility and maintain the overall system efficiency.
- Example of a low-power readout circuit: We have included a reference to a readout circuit implemented in [14], which demonstrates the feasibility of designing a low-power readout circuit. This circuit operates with a 0.5 V drive voltage and consumes only 3.95 uW of power while achieving a low noise of 45 nV/√Hz and a bandwidth of 408 kHz, which helps minimize phase error.
- THD performance comparison: We have noted that the THD performance of the input current wave in [14] was 0.66%, which is higher than the current wave THD achieved by our CG. This highlights the importance of designing the readout circuit with high linearity to avoid compromising the overall linearity of the generated current.
- Future readout circuit design considerations: We have emphasized that the most critical aspect of the readout circuit we plan to design in the future is to achieve a THD of less than 0.04% when inputting a pure sine wave that satisfies the targeted input amplitude. This stringent THD requirement must be met while simultaneously satisfying the low power and low area constraints.
These revisions, located on pages L. 351-372
Comments 5 : The authors claim that the BIS IC can be applied in various fields, including medical diagnostics and body composition analysis. A more detailed discussion on the specific implementation of the design in these applications would strengthen the paper, providing clear insights into its practical uses.
Response 5 : Thank you for your suggestion to provide a more detailed discussion on the specific implementation of our proposed BIS IC in various applications. We agree that elaborating on the practical uses of our design will strengthen the paper and offer clearer insights into its potential impact in fields such as medical diagnostics and body composition analysis.
In the revised manuscript, we have expanded on the discussion of the BIS IC's applications and its advantages over existing commercial electrical impedance spectroscopy (EIS) systems. Specifically, we have made the following additions:
- Comparison with commercial EIS systems: We have highlighted that commercial EIS systems, such as [18], which employ single-tone excitation methods, can measure a wide range of impedances with high accuracy. However, these systems often have limitations in terms of portability and power requirements. They tend to be large in size, making them difficult to carry, and require an external power supply due to their high power consumption.
- Advantages of our proposed CG IC: In contrast, our proposed CG IC offers several key advantages that make it suitable for portable and insertable devices. With a small area of 0.0798 mm2 and low power consumption of 19.2 uW, our design enables the development of compact and energy-efficient BIS applications. Moreover, the frequency tunability of our CG IC allows for the acquisition of a wide range of information, enhancing its versatility in various applications.
- Potential applications in medical diagnostics and body composition analysis: We have elaborated on the potential applications of our BIS IC in medical diagnostics and body composition analysis. In medical diagnostics, our design can be integrated into portable devices for point-of-care testing, enabling rapid and accurate assessment of various physiological parameters. For body composition analysis, the small size and low power consumption of our CG IC make it suitable for wearable or implantable devices that can continuously monitor body composition changes over time.
By providing these additional details and comparisons, we aim to highlight the practical advantages of our proposed BIS IC and its potential for implementation in various applications. The small area, low power consumption, high linearity, and frequency tunability of our design make it particularly well-suited for portable and insertable devices, opening up new possibilities in fields such as medical diagnostics and body composition analysis.
These revisions, located in lines L. 373-404 of the manuscript, address your concerns and provide a clearer picture of the specific implementation and practical uses of our BIS IC.

Round 2
Reviewer 2 Report
Comments and Suggestions for Authors
Authors have revised the manuscript, adding sufficient details to clarify the figures and principles. The revised version looks good.